# Cryo-EM structure of the polycystin 2-l1 ion channel

Raymond E Hulse[1,2], Zongli Li[3], Rick K Huang[1], Jin Zhang[1,4], David E Clapham[1,2]*

[1]Janelia Research Campus, Howard Hughes Medical Institute, Ashburn, United States; [2]Department of Cardiology, Howard Hughes Medical Institute, Boston Children's Hospital, Boston, United States; [3]Department of Biological Chemistry and Molecular Pharmacology, Howard Hughes Medical Institute, Harvard Medical School, Boston, United States; [4]School of Basic Medical Sciences, Nanchang University, Nanchang, China

**Abstract** We report the near atomic resolution (3.3 Å) of the human polycystic kidney disease 2-like 1 (polycystin 2-l1) ion channel. Encoded by PKD2L1, polycystin 2-l1 is a calcium and monovalent cation-permeant ion channel in primary cilia and plasma membranes. The related primary cilium-specific polycystin-2 protein, encoded by PKD2, shares a high degree of sequence similarity, yet has distinct permeability characteristics. Here we show that these differences are reflected in the architecture of polycystin 2-l1.

DOI: https://doi.org/10.7554/eLife.36931.001

## Introduction

Revolutionary improvements in resolving protein and cellular structures, and genetic identification of ciliopathies, have created renewed interest in primary cilia. These small protuberances (5–10 µm in length) are found in nearly every cell type. Primary cilia house key downstream elements of the sonic hedgehog pathway, which regulates embryonic development and some cancers. Recently, new tools have enabled measurements of other signaling elements within cilia. Primary cilia have elevated resting internal $Ca^{2+}$ concentrations ([$Ca^{2+}$]=300–700 nM [*Delling et al., 2013*]), and possess at least two TRP channels, polycystin-2 and polycystin 2-l1. These ion channels are encoded by the PKD2 and PKD2L1 genes of the TRPP subfamily. Electrophysiological measurements demonstrate that polycystin 2-l1 and polycystin-2 both underlie ionic currents in primary cilia (*DeCaen et al., 2013*, *2016*; *Kleene and Kleene, 2017*). This suggests that voltage gradients and $Ca^{2+}$, in addition to cAMP, are relevant signals within primary cilia.

PKD2 and PKD2L1 share high degrees of sequence identity (52%) and similarity (71%). Mutations in PKD2 account for ~15% of cases of individuals afflicted with Autosomal Dominant Polycystic Kidney Disease (ADPKD). In contrast, there are no diseases currently linked to PKD2L1. Deletion of PKD2 is lethal in mice (*Wu et al., 1998*), but deletion of PKD2L1 results in occasional gut malrotation, a relatively mild *situs inversus* phenotype (*Delling et al., 2013*). Finally, while the functional characterization of polycystin 2-l1 protein has been enabled by its plasma membrane expression (*DeCaen et al., 2013*, *2016*), polycystin-2 protein function has been hampered by its restriction to surface expression in primary cilia. Recent approaches have been successful, however, in establishing the key biophysical properties of polycystin-2. These approaches include a gain-of-function mutation that enables expression to the plasma membrane in *Xenopus* oocytes (*Arif Pavel et al., 2016*) and direct patch clamp recording of polycystin-2 in primary cilia (*Kleene and Kleene, 2017*; *Liu et al., 2018*). These studies establish polycystin-2 as a monovalent-selective cation channel with little or no calcium permeation. In contrast, polycystin 2-l1 shows significant and relevant $Ca^{2+}$ permeability

*For correspondence:
claphamd@hhmi.org

Competing interests: The authors declare that no competing interests exist.

($P_{Ca}/P_{Na}$ ~6) (*DeCaen et al., 2013*, *2016*). Despite this knowledge, the physiological stimulus for activation and gating is unclear for both channels.

In a similar fashion, structural information about membrane proteins has undergone a profound advance. This is largely due to the explosion of structures from single particle electron cryo-microscopy (cryo-EM) in recent years. Accordingly, multiple structures have been solved for the TRP ion channel family, including the TRPP family to which polycystin-2 and polycystin 2-l1 belong.

Here we present a structure of the full-length human polycystin 2-l1 protein at 3.3 Å resolution in cryo-EM. The overall architecture conforms to that of other TRP channels, and in particular to the TRPP polycystin-2 structures. We establish the core structure and point out differences in regions and residues that may account for polycystin-2 and polycystin 2-l1's distinct permeation and gating.

## Results

We first expressed and purified full-length recombinant polycystin 2-l1 protein tagged with maltose binding protein (MBP) (*Figure 1—figure supplement 1*) using HEK293 GnT I⁻ cells and the BacMam system (*Goehring et al., 2014*). To determine whether the MBP-tagged protein was functional, we measured single channel activity in reconstituted liposomes under voltage clamp. The channel's conductance in the same buffer used for purification (HKN, see Materials and methods) was 105 pS, consistent with previous single-channel studies from cells expressing polycystin 2-l1 (*DeCaen et al., 2013*) (*Figure 1—figure supplement 1d*).

### General architecture of polycystin 2-l1

Polycystin 2-l1's structure exhibits many of the hallmarks of TRP channels: it forms a homotetramer with a domain-swapped Voltage Sensing-Like Domain (VSLD, the S1-S4 transmembrane domains) (*Figure 1a*) and shares remarkable architectural similarity to polycystin-2 (*Figure 1—figure supplement 2*) (RMSD of 1.5 Å measured with polycystin-2 model pdb 5T4D). As characteristic of group 2 TRP channels (TRPPs and TRPMLs), polycystin 2-l1 has a long S1-S2 extracellular loop, termed the polycystin mucolipin domain (PMD). This domain, a series of 3 α-helices, 4 β-sheets, and a glycosylated three loop region (see below, TLC or three-leaf clover), forms a cover, or lid, above the channel (*Figure 1b*). In contrast to the 3 or 4 observed glycans in polycystin-2's polycystin mucolipin domain (PMD) (*Grieben et al., 2017*; *Shen et al., 2016*; *Wilkes et al., 2017*), polycystin 2-l1's PMD has one clear glycan density located at residue N207. An additional density, at N241, could possibly support an additional glycan. Notably, the PMD interacts with underlying elements of the pore, the voltage sensing-like domains, and adjacent PMDs. The role of glycosylation of the PMD in TRPPs is unknown. Two possibilities include serving as a folding/quality control mechanism or as a site for intermolecular interactions (e.g., ligand involved in gating).

The intracellular face of polycystin 2-l1 is too poorly resolved to model the N terminus and the S4-S5 linker densities. Also, despite expression of full-length protein (*Figure 1—figure supplement 1*) the C terminus of polycystin 2-l1 was not resolved. We conclude that these elements are either unstructured or connected with flexible regions, which, along with the lack of cytoplasmic elements in 2D classification in polycystin-2 (*Shen et al., 2016*), prevents model building of these regions.

### The voltage sensing-like domain

The VSLD is comprised of four transmembrane helical spanning elements that, while similar in structure to the voltage-sensing domains of voltage-gated ion channels, do not convert the energy of the transmembrane electric field into pore gating. Indeed, none of the group II TRP channels have significant voltage dependence. Comparison of the VSLD of polycystin 2-l1 and polycystin-2 reveals that the polycystin 2-l1 S2 helix is tilted an additional 4.5° away from the core, is laterally shifted away from the rest of the VSLD, and shows the greatest local RMSD difference (*Figure 1—figure supplement 2a, b and c*). Polycystin 2-l1's S3 is a near-continuous helix extending from the membrane-spanning region into a pocket created by the PMD (*Figures 1c* and 3b). This S3 helix exhibits greater secondary structure than the presumed closed polycystin-2 structure of Shen et al. (*Shen et al., 2016*), but is similar to that of the multiple- and single-ion models of Wilkes et al., (*Wilkes et al., 2017*). The S3 extended helix abuts the PMD of the same monomer but does not display the same cation-π interactions as polycystin-2's F545 and R320 residues. The S3-S4 linker and the top portion of the S4 helix also fit into the same cleft of the PMD as in polycystin-2. In

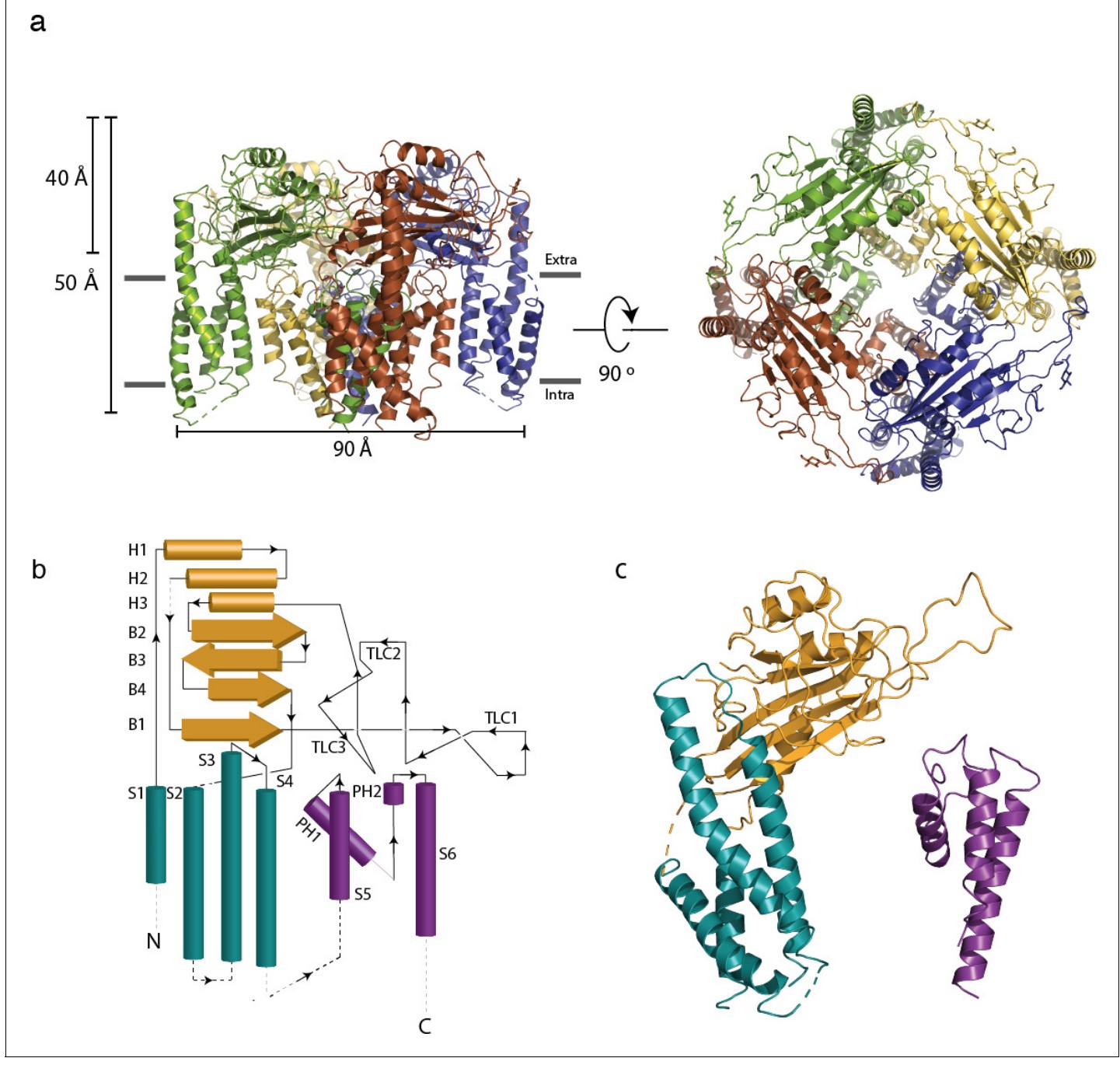

**Figure 1.** Architecture of polycystin 2-l1 tetrameric ion channel. (**a**) Side view parallel to the membrane and from the top (extracellular) surface. Distinct subunits in the tetramer are color-coded. (**b**) 2D topological representation of the polycystin 2-l1 monomer; voltage sensor-like domain (VSLD, teal), polycystin mucolipin domain (PMD, orange), and pore domain (PD, violet). (**c**) Monomer color-coded and matched to *Figure 1b*: N terminus/VSLD (teal), PMD (orange), C terminus/PD (violet).

DOI: https://doi.org/10.7554/eLife.36931.002

The following figure supplements are available for figure 1:

**Figure supplement 1.** Purification, negative staining, 2D class average and function of polycystin 2-l1.

DOI: https://doi.org/10.7554/eLife.36931.003

**Figure supplement 2.** Comparison of polycystin 2-l1 and polycystin-2.

DOI: https://doi.org/10.7554/eLife.36931.004

**Figure supplement 3.** Single particle data processing.

*Figure 1 continued on next page*

*Figure 1 continued*

DOI: https://doi.org/10.7554/eLife.36931.005

**Figure supplement 4.** 2D and 3D Fourier Shell Correlation (FSC) curve analysis and local resolution map of polycystin 2-l1.

DOI: https://doi.org/10.7554/eLife.36931.006

**Figure supplement 5.** Quality of the EM density map and fit of model.

DOI: https://doi.org/10.7554/eLife.36931.007

polycystin-2, the S4 adopts a $3_{10}$-helix configuration (I571 – F579). The density of the S4 helix of polycystin 2-l1 is incomplete at this region so a $3_{10}$ configuration is not observed. Similarly, the equivalent residues of polycystin-2 that are thought to form a salt bridge between the S3 and S4 helix (K572, K575) (*Shen et al., 2016*), do not have sufficient side chain density to model accurately. The equivalent residue for polycystin-2's acidic D511 residue is D390 in polycystin 2-l1, which does not form a salt-bridge with neighboring residues.

## The polycystin mucolipin domain

The polycystin mucolipin domain (PMD) rests on top of the VSLD of the same monomer and the adjacent pore domain (*Figure 3b and c*), with a series of three α-helices facing inward towards the funnel/turret of the pore (*Figure 3c*). A series of β-sheets is sandwiched between these α-helices with several loops that interface with the extracellular space and adjacent PMD (*Figures 1b* and *3c*). Finally, a disulfide bond is present between residues C210-C223 in the PMD, similar to several polycystin-2 structures (*Figure 3d*). This element is proposed to stabilize a loop in polycystin-2 (*Shen et al., 2016*; *Wilkes et al., 2017*) and seems likely to play the same role in polycystin 2-l1.

### Three leaf clovers (TLCs) of the polycystin mucolipin domain (PMD)

Grieben et al., (*Grieben et al., 2017*) described the three-lobed area of polycystin-2's PMD, naming it the 'three-leafed clover'. The same area is found in polycystin 2-l1, but with several differences. Polycystin 2-l1 lacks the small α-helix in TLC1 of polycystin-2. Interestingly, this region (D208-D225) displays moderately lower sequence conservation among the TRPPs (*Figure 3a*). As noted, TLC1 in polycystin-2 appears to extend into the adjacent PMD and interact with the S3 helix and S3-S4 linker (*Grieben et al., 2017*; *Wilkes et al., 2017*). In polycystin 2-l1, the analogous TLC1 extends from one monomer into the PMD of the adjacent subunit (*Figure 3b*). However, the nature of the interaction differs in that F216 of TLC1 is near Y308 of the adjacent subunit's PMD, representing a possible π - π stacking interaction or a hydrophobic pocket (*Figure 4a*). Additionally, N311's carboxamide oxygen group, located on a hairpin between β3 and β4 of the PMD, is within hydrogen bonding distance (3.3 Å) of Y224's amide in TLC1 of the adjacent subunit of polycystin 2-l1 (*Figure 4b*). Finally, residue W259 of TLC3 formats a cation-π stacking interaction with the upper pore domain at residue R534 (*Figure 4c*). We interpret these interactions as supporting tetrameric assembly and stability.

### Fenestrations

Despite moderately low identity and an increase in charged residues in polycystin 2-l1's TLC3 (SPDKEE (residues 228–232) versus SVSSED in polycystin-2), the loop appears essentially the same as in polycystin-2. This element appears to reach under the PMD's β-sheet. Proximity of the returning loop of TLC1 towards TLC2, as well as the S5-PH1 loop of the monomer and the PH2-S6 loop adjacent monomer creates four lateral openings at the base of the PMD in polycystin 2-l1, as in polycystin-2. These could present an alternative route for ion permeation (*Grieben et al., 2017*; *Wilkes et al., 2017*). This area is less conserved among the polycystin 2-l1 homologs (PKD2, PKD2L2)(*Figure 3a*).

Polycystin 2-l1's PMD also interacts with the adjacent subunit's upper pore domain. In the polycystin-2 multiple- (pdb 5MKF) and single-ion (pdb 5MKE) structures, Wilkes et al., (*Wilkes et al., 2017*) observed glycosylation-dependent interactions of the PMD of one subunit with the loop pore-helices of the adjacent (*Wilkes et al., 2017*), with mutually exclusive glycosylation states. One last interaction between TLC3 and the loop between pore-helix 2 and S6, noted in the polycystin-2 structure (*Shen et al., 2016*), is seen in polycystin 2-l1: W259, from TLC3, forms a cation-π stacking interaction with R534 and a hydrogen bond with G260's carbonyl (2.8 Å) (*Figure 4c*). This difference may

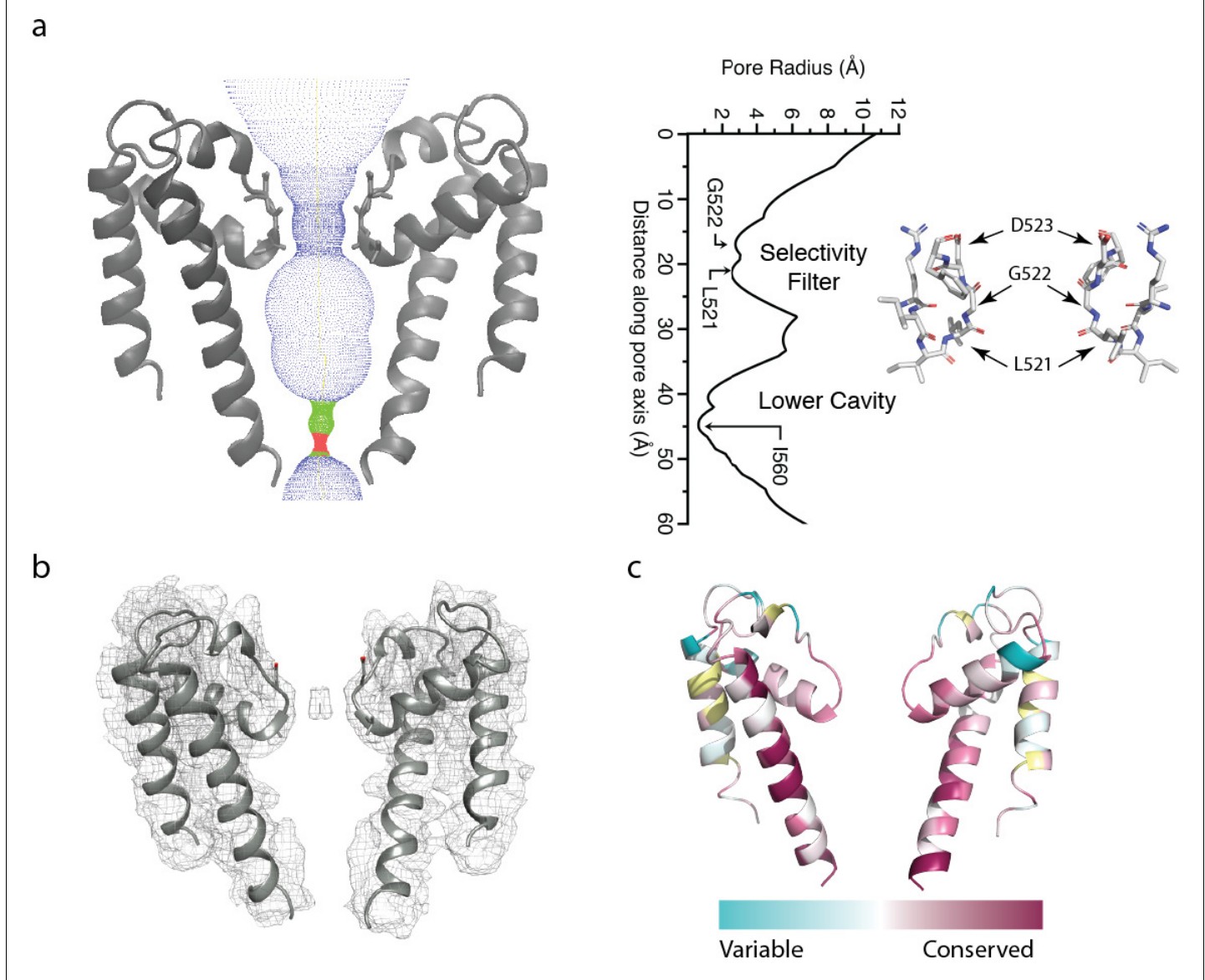

**Figure 2.** The polycystin 2-l1 pore domain. (**a**) Path of permeation depicted with HOLE (left). At right is a close-up view of the selectivity filter (two monomers removed for clarity) with three conserved residues L521, G522, and D523. Pore radius (HOLE) measured along the pore axis (left). (**b**) Electron density map superimposed on the polycystin 2-l1 model; contour level 5.0. (**c**) TRPP family multiple sequence alignment of conservation projected in color onto the polycystin 2-l1 pore domain. Two of four monomers have been removed to increased clarity.

DOI: https://doi.org/10.7554/eLife.36931.008

help rationalize the intermediate pore size observed in the polycystin-2 multiple-ion structure (1.4 Å) (*Wilkes et al., 2017*) and the single-ion structure (1.0 Å) (*Wilkes et al., 2017*) where such interactions do not exist, and our polycystin 2-l1 structure (2.6 Å).

## The pore domain

As in polycystin-2, polycystin 2-l1's selectivity filter is flanked by two pore-helices, which are themselves each flanked by a transmembrane spanning helix (S5 and S6). The S6 helix of polycystin 2-l1 remains α-helical throughout its length, whereas the S6 helix of polycystin-2 is broken by a middle π-helix element, before continuing to the C terminus (*Shen et al., 2016*) (*Figure 1—figure supplement 2b and c*). The S6 helix of polycystin 2-l1 is also tilted 7.9° relative to polycystin-2 (*Figure 1—figure supplement 2c*). The key filter residues of polycystin 2-l1 are L521, G522 and D523, in which

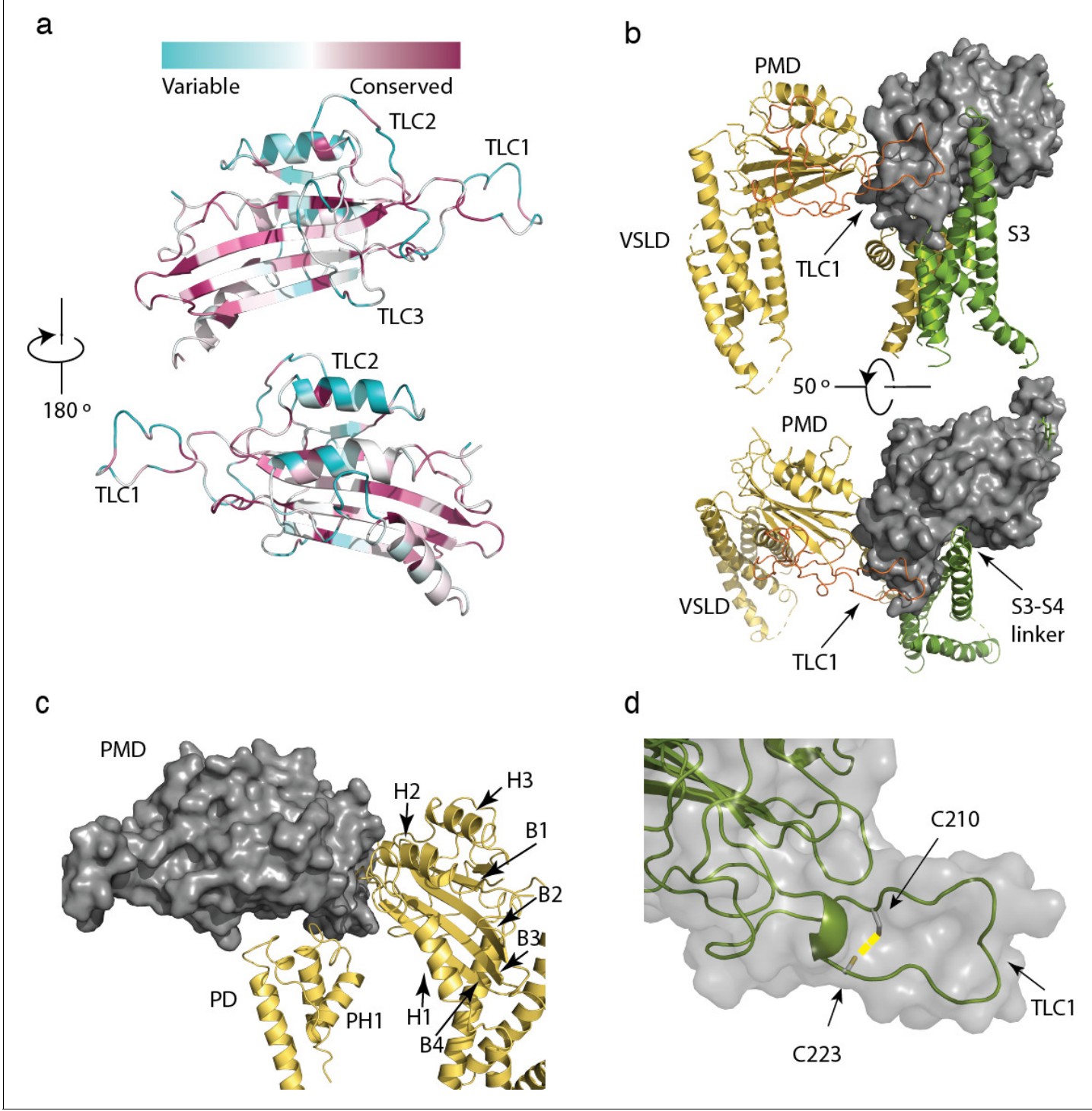

**Figure 3.** Structure and conservation of the polycystin mucolipin domain (PMD) of polycystin 2-l1. (a) sequence conservation projected in color onto the PMD domain for polycystin 2-l1 homologs; Three-leaf clover (TLC) domains labeled. (b) The S3 (green), S3-S4 linker (green, rotated 50°), and TLC1 from the adjacent PMD (red) interact with the PMD of the same domain (gray). (c) Interactions of the PMD (gray) with the pore domain (PD) and PMD of the adjacent monomer (yellow). PMD secondary structural elements helices 1–3, and β-sheets 1 to 4 labeled, along with the PD and Pore Helix (PH1). (d) Disulfide C210-C223 (yellow) in the TLC1 loop (3.4 Å) of the PMD.

DOI: https://doi.org/10.7554/eLife.36931.009

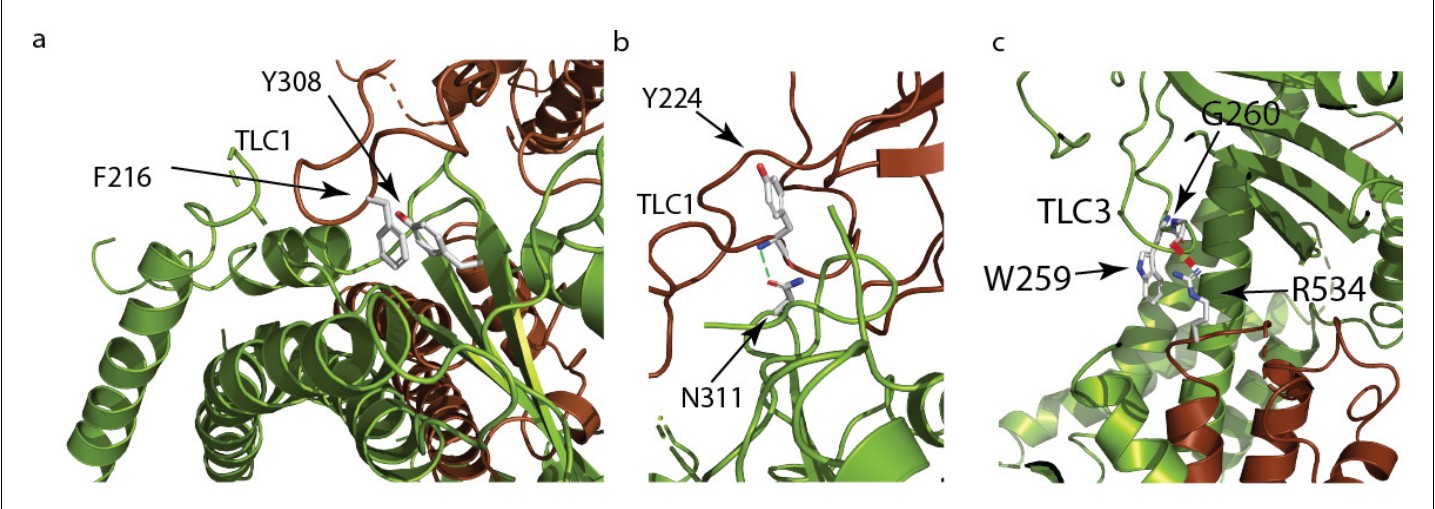

**Figure 4.** Polycystin mucolipid domain interactions of polycystin 2-l1. (**a**) π-π interaction of residue residing in TLC1 F216 (brown) with the neighboring (green) PMD (Y308). (**b**) Hydrogen bonding interaction (green dash) of PMD residue N311 with the neighboring PMD residue Y224's amide (blue). (**c**) Residue W259 in TLC3 of the PMD forms a cation-π stacking interaction with the upper pore domain residue R534. The guanidino group of R534 also forms a hydrogen bond (red dash) with the carbonyl of G260 of the adjacent PMD domain.
DOI: https://doi.org/10.7554/eLife.36931.010

the carbonyls of L521 and G522 point to the central pore axis (*Figure 2a*). D523's side chain is well-resolved; the side chains face upward in parallel to the axis of ion conduction. The pore domain shows moderately strong conservation in S5, S6, and elements of the selectivity filter, as well as in pore helix 1 (*Figure 2c*). Notably, however, K511 in polycystin 2-l1 is highly variable among equivalent residues in other TRPPs (*Figure 2c*). The corresponding residue is a negatively-charged glutamate in polycystin-2 and an asparagine in polycystin 2-l2. The effect of this residue is to confer a net positive electrostatic potential (*Figure 5a*) relative to polycystin-2's net negative charge.

Relative to polycystin-2's pore helix 1 (PDB 5T4D), the angle between the beginning of pore helix 1 and the top of the selectivity filter is similar (77° for polycystin 2-l1 and 75° for polycystin-2) compared to the 10° pitch difference between polycystin-2 and TRPV1 (*Grieben et al., 2017*). However, polycystin 2-l1's pore helix, measured from beginning to end of the α-helix at the Cα, is ~1.5 Å shorter than that of polycystin-2.

The narrowest apertures in the selectivity filter, measured by HOLE (*Smart et al., 1996*), have radii of 2.6 and 2.8 Å at L521 and G522 respectively (*Figure 2a*). These distances represent a larger opening than that in any of the three polycystin-2 structures (1.0 to 1.4 Å) (*Grieben et al., 2017*; *Shen et al., 2016*; *Wilkes et al., 2017*). Such a diameter is sufficient to accommodate a partially hydrated $Ca^{2+}$ ion with an average ~2.4 Å metal-oxygen distance (*Katz et al., 1996*; *Marcus, 1988*) and an ionic radius of 0.99 Å (*Pauling, 1988*; *Hille, 2001*). An observed density in the selectivity filter extends across the two carbonyls of L521 and G522, contrasting with polycystin-2 densities that localize above and below the selectivity filter (*Wilkes et al., 2017*) (*Figure 2b*). However, cryo-EM cannot determine the ion's identity with the same degree of reliability of X-ray anomalous scattering. Although defined buffers are used, we cannot rule out the possibility of less abundant ions occupying the site.

## Ion permeation differences between polycystin-2 and polycystin 2-l1

Polycystin-2 and polycystin 2-l1 core selectivity filters residues (LGD) are conserved, leading us to examine other explanations as to why polycystin 2-l1 conducts $Ca^{2+}$ while polycystin-2 does not ($P_{Ca}/P_{Na}$ ~6) (*DeCaen et al., 2013, 2016*; *Kleene and Kleene, 2017*; *Liu et al., 2018*). If the pore helix 1-selectivity filter-pore helix 2 region of polycystin 2-l1 (C512-P538) is substituted by polycystin-2's analogous region (C632-P658), the polycystin 2-l1 chimera has roughly (within 4-fold) similar permeability to $Na^+$, $K^+$, and $Ca^{2+}$ (*Shen et al., 2016*). Point mutation experiments in which polycystin 2-l1's D523 and D525 were mutated to alanine or serine yielded no measurable currents

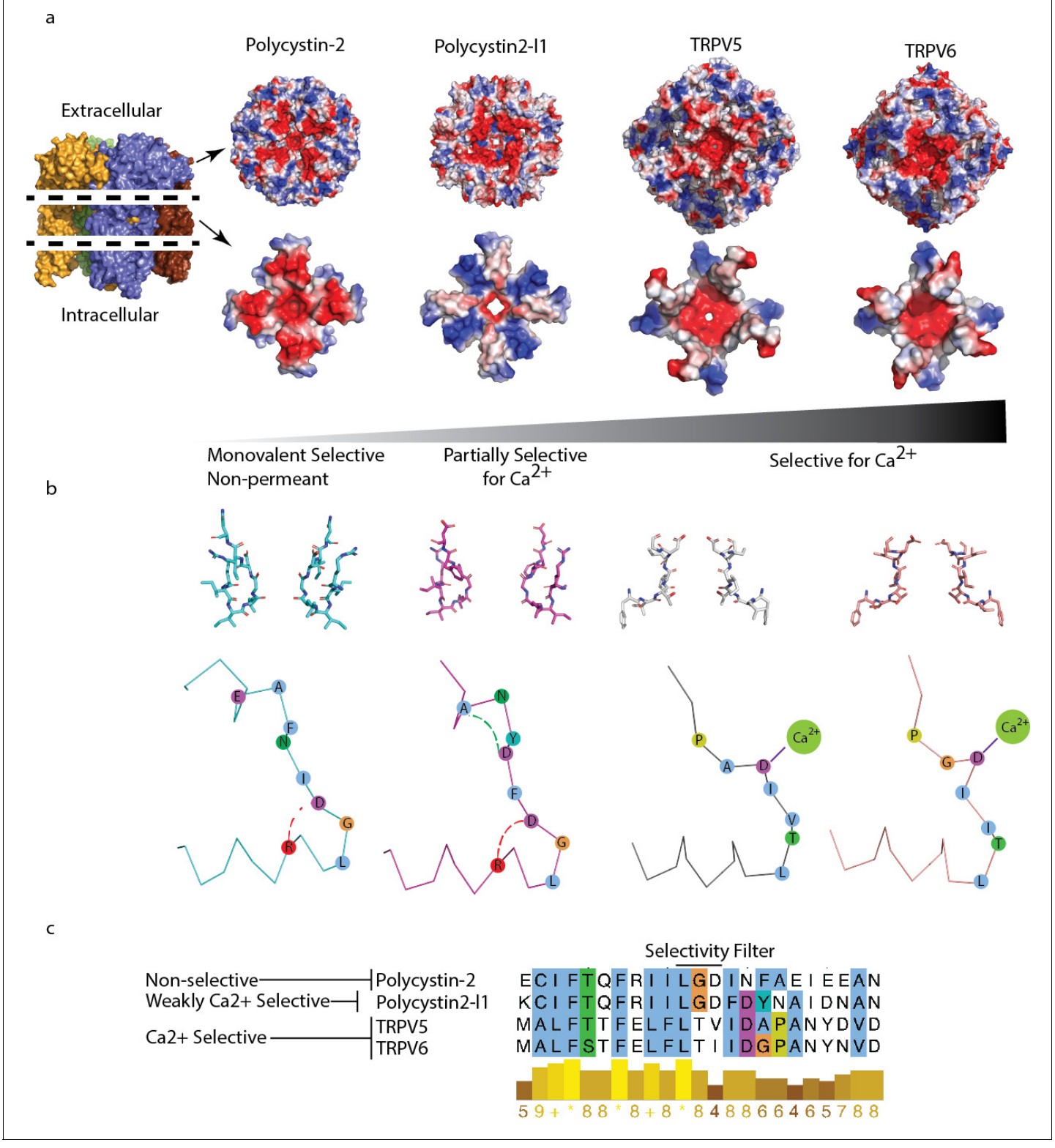

**Figure 5.** Comparison of charge landscape and local bonding between TRPP and the calcium-selective TRPV5 and TRPV6 channels. (a) The extracellular facing (top) and selectivity filter region (below) of channels with progressively increasing relative permeability to $Ca^{2+}$; cut-away schematic (far left) for orientation. (b) Stick representation of the selectivity filter region (top) and simplified diagram (below). Local bonding characteristics are highlighted: conserved salt-bridge (red) interaction of polycystin-2 and polycystin 2-l1, the unique hydrogen bond of polycystin 2-l1 (green), and the direct side-chain

*Figure 5 continued on next page*

*Figure 5 continued*

coordination of $Ca^{2+}$ in TRPV5 and TRP6. (**c**) Alignment with conservation scoring and clustalW coloring scheme of the selectivity filter region show in (**b**).

DOI: https://doi.org/10.7554/eLife.36931.011

(*DeCaen et al., 2013*). However, polycystin 2-l1(LG**D**$_{523}$: D523N) was 18-fold, and polycystin 2-l1 (LGDF**D**$_{525}$: D525N) 9-fold less calcium-permeant ($P_{Ca}{}^{2+}/P_{Cs}{}^{+}$) than wt polycystin 2-l1. Also, and significantly, the D525N mutation reduced outward current block by $Ca^{2+}$, while D523N did not (*DeCaen et al., 2016*). These experiments indicate that order and placement of negative charge in the narrow region of the pore is important for both $Ca^{2+}$ permeation and block. Higher resolution cryo-EM structures that identify $Ca^{2+}$ and/or monovalents that are occupying the pore with high certainty (or equivalent crystallographic studies like those of CaVAb [*Tang et al., 2014*]), and molecular dynamic simulations, should shed light on these issues.

As a final attempt to understand the differences in the relative selectivity of TRP channels, we compared the charge landscape of the pore helix-selectivity filter-pore helix assembly of polycystin-2 and polycystin 2-l1. When the PMD is stripped away and only the pore-helices and selectivity filter elements of both channels are visualized, a striking difference appears (*Figure 5a*). Polycystin 2-l1 has a more restricted negative charge density immediately surrounding the pore than does polycystin-2. Interestingly, its pore-helices form a 'Maltese cross' of net positive charge, rotated 45° to the negative charge distribution 'cross' in polycystin-2. We compare these charge distributions to the much more $Ca^{2+}$-selective TRPV5 and TRPV6 ($P_{Ca}{}^{2+}/P_{Na}{}^{+} > 100$) (*Ramsey et al., 2006*) structures (*Hughes et al., 2018*; *McGoldrick et al., 2018*) (*Figure 5a*) in which the TRPV5/6 channels' negative charge is much more localized around their respective pores.

An additional mechanism to explain different selectivities of TRPPs is a local bonding environment difference at and near the selectivity filter between polycystin-2 and polycystin 2-l1. A conserved salt bridge between D523 and R518 in polycystin 2-l1 exists in polycystin-2 (D643 and R638). Yet a hydrogen bond between residues D525 and A528 in polycystin 2-l1 is not present in polycystin-2; the residues are not conserved (*Figure 5b and c*). Local hydrogen bonding networks at the selectivity filter of ion channels have been observed to influence function such as the entry into a C-inactivated state for KcsA (*Cordero-Morales et al., 2011*), but such details for non-selective channels are currently unknown.

In both observations, we rely on one structure, which may change with gating. The use of molecular dynamics (MD) to observe the simulated trajectory of ions and their relationship to the energy landscapes of the selectivity filter is an excellent starting point for future experiments. Coupling future structures with functional mutagenic studies (e.g., the weakly conserved residue K511) should answer these questions.

## Discussion

Although polycystin 2-l1 and polycystin-2 structures are very similar, there are notable differences that may explain reported functional differences in their selectivity and permeation. For discussion, we divide polycystin 2-l1 into three regions to highlight these differences; the voltage sensing-like domain (VSLD), the pore mucolipin domain (PMD), and the pore (PD). In the VSLD, polycystin 2-l1's S3 helix extends into the PMD as a complete α-helix. In addition, the top of the S3-S4 helical region abuts the PMD. Comparing the PMDs, polycystin 2-l1 lacks oligosaccharides that act as bridges between adjacent subunits. Finally, although similar, polycystin 2-l1's pore diameter is slightly larger.

Due to our inchoate understanding of the TRPP family, we must be cautious in linking the observed differences in structure to functional differences. The two main features we seek to understand are selectivity and gating. Comparing selectivities of the two channels, we know that polycystin 2-l1 conducts both monovalent ions and $Ca^{2+}$, while polycystin-2 is monovalent-selective (*Kleene and Kleene, 2017*; *Liu et al., 2018*). We explain this difference with two features: pore diameter and electrostatic fields. The relatively larger size of polycystin 2-l1's pore radius (2.6–2.8 Å) may enable partially hydrated calcium's transit, thus avoiding the larger energy required for pore residues to dehydrate calcium. Most striking are the unique electrostatic maps (*Figure 5a*) surrounding the polycystin-2 and polycystin 2-l1 pores. Such differences in charge suggest that the putative

energy landscapes may create small variations in selectivity, a characteristic suggested for the prokaryotic NaK ion channel (*Alam and Jiang, 2011*). These slight charge differences, and so landscapes, may impart the subtle relative permeability differences observed among most TRP channels. For organelles in which the TRPP (cilia) and TRPML (endolysosomes) appear, these subtle differences may have driven the evolution of small changes in relative permeation in order to titrate levels of ions within these sub-femtoliter compartments. Research into the relative permeabilities of the related TRPML family of ion channels will further our understanding of mechanisms beyond direct side chain coordination.

TRPPs are not appreciably voltage- or mechanically-gated within physiologically relevant ranges of these forces (*DeCaen et al., 2013*, *2016*), which raises the question of what other mechanisms activate these channels. The large surface area of the PMD is a logical candidate for binding of ligands. An appealing hypothesis is that unknown ligands bind the PMD, initiates an interaction with the VSLD (possibly at the extended S3 helix) and/or pore domain which alters channel function. Similarly, interactions of the PMD with the pore domain may elicit conformational changes leading to gating. Given the association of polycystin-1 (a protein of unknown function) and polycystin-2 proteins, it will be important to determine if any of the domains of polycystin-1 proteins interact with polycystin-2 family PMDs.

PI(4,5)P$_2$ was recently observed to facilitate the gating of both polycystin-2 and polycystin 2-l1 by interacting with residues on both the N and C termini (*Zheng et al., 2018*). The current lack of resolution of N- and C-terminal ion channel flexible regions limits our interpretation of gating influences for the group II TRP channels. Among group I TRP channels, the large interaction surfaces for intracellular ligands, such as the ankyrin repeats found on several TRPs, the more common TRP domains, and predicted C-terminal EF-hand calcium-binding sites and calmodulin-binding domains, are difficult to interpret without better understanding of proteins or ligands that may bind these regions.

The explosion of structures available for TRP channels lays the groundwork for interpreting future detailed characterization. The structure of polycystin 2-l1 presented here, will help in understanding what drove evolution to create the diversity that exists in the TRPP subfamily, but only when we more completely understand their biophysical and physiological functions. The recent progress made in cryo-EM underscores the importance of discovering these features.

# Materials and methods

**Key resources table**

| Reagent type (species) or resource | Designation | Source or reference | Identifiers | Additional information |
|---|---|---|---|---|
| Gene (*Homo sapiens*) | PKD2L1 | Synthetic, non-codon optimized | Uniprot - Q9P0L9 | |
| Cell line (*Homo sapiens*) | HEK 293 GnT I- | ATCC | ATCC: CRL-3022/ RRID: CVCL_A785 | |
| Cell line (*Spodoptera frugiperda*) | Sf9 | ATCC | ATCC: CRL-1711/RRID: CVCL_0549 | |
| Recombinant DNA reagent | pEG BACMAM | doi: 10.1038/nprot.20.14.173 | | |
| Software, algorithm | cisTEM | doi: 10.7554/eLife.35383 | | http://cistem.org |
| Software, algorithm | Pymol | PyMOL Molecular Graphics System, Schrödinger, LLC | RRID:SCR_000305 | http://www.pymol.org |
| Software, algorithm | UCSF Chimera | UCSF Resource for Biocomputing, Visualization, and Bioinformatics | RRID:SCR_004097 | |
| Software, algorithm | 3D FSC | doi: 10.1038/nmeth.4347 | | https://3dfsc.salk.edu/ |
| Software, algorithm | PHENIX | doi.org/10.1107/S0907444909052925 | RRID:SCR_014224 | https://www.phenix-online.org |
| Software, algorithm | Coot | doi.org:10.1107/S0907444910007493 | RRID:SCR_014222 | http://www2.mrc-lmb.cam.ac.uk/ personal/pemsley/coot/ |
| Software, algorithm | HOLE | (see References) | | http://www.holeprogram.org |

*Continued on next page*

*Continued*

| Reagent type (species) or resource | Designation | Source or reference | Identifiers | Additional information |
|---|---|---|---|---|
| Software, algorithm | VMD | UIUC Theoretical and Computational Biophysics Group | | http://www.ks.uiuc.edu/Research/vmd/ |

## Cloning, expression and purification

Cloning, expression, and purification of full-length human polycystin 2-l1 were completed using the BacMam strategy (*Goehring et al., 2014*). Briefly, constructs were subcloned into the vector pEG using restriction sites NotI and BstBI. The construct sequence was verified and transformed into DH10bac cells. Blue-white screening facilitated colony selection to create a midiprep of the bacmid. This preparation was used to transfect Sf9 cells using Cellfectin II following the manufacturer's instructions. Amplification of the virus was completed for 2 cycles (P2) before viral particle were used to infect HEK293S GnT I$^-$ cells at 10% v/v at 37°C and 8% $CO_2$ with shaking. After 24 hr, sodium butyrate was added to a final concentration of 10 mM and cells were harvested after 48 hr.

Protein was extracted directly from pellets using 40x critical micellar concentration (CMC) of the detergent C12E9 in HKN buffer (in mM: HEPES 50, KCl 150, NaCl 50, $CaCl_2$ 5; pH 7.5) with an EDTA-free protease inhibitor cocktail (Roche) tablet for 2 hr at 4°C using gentle rotation. Ultracentrifugation of the sample was completed at 40,000 RPM for 1 hr at 4°C. The resulting supernatant was retained and used for a batch incubation with amylose resin overnight at 4°C. Resin was collected, washed in HKN buffer with higher K$^+$ (500 mM) and 2x CMC of *n*-Dodecyl β-D-maltoside: cholesteryl hemisuccinate 10:1 (DDM:CHS) for 10 bed volumes followed by a normal HKN buffer for 10 bed volumes with 2xCMC of DDM:CHS. The protein was eluted using 40 mM maltose in HKN buffer at 2xCMC DDM: CHS. All fractions were collected, concentrated and subjected to 1 round of size exclusion chromatography using an Increase Superose6 (GE Lifesciences) column pre-equilibrated with HKN and 2xCMC DDH: CHS. Protein-containing fractions, as determined by A280 and western-blotting against maltose binding protein (MBP), were collected, concentrated and used for further preparation. The sample's concentration was again determined using A280 and incubated with a 1:3 ratio (mass) of poly (maleic andydride-alt-1-decene substituted with 3-(dimethylamino) propylamine; PMAL C8) overnight with gentle rotation at 4°C. A final round of size exclusion chromatography using an Increase Superose6 column equilibrated with HKN buffer was performed, fractions collected, and concentrated using a Vivaspin Turbo4 100,000 molecular weight cutoff (MWCO) centrifugation device before being applied to grids and freezing.

## Sample preparation

The cryo specimen was vitrified using 3 µl of purified polycystin 2-l1 in PMAL-C8 at 3.5 mg/ml and applied onto a glow-discharged 400 mesh copper Quantifoil R1.2/1.3 holey carbon grid (Quantifoil). Grids were blotted for 3 s at 95% humidity and flash frozen in liquid nitrogen-cooled liquid ethane bath using a Vitrobot (Thermo Fisher, Hillsboro OR). Samples were stored in liquid nitrogen until acquisition.

## Cryo-EM image acquisition, processing and modeling

Data for full-length polycystin 2-l1 were collected on a FEI Titan Krios (Thermo Fisher, Hillsboro, OR) at 300 kV with a Gatan Quantum Image Filter (20 eV slit) and K2 Summit detector at the Janelia Research Campus (Ashburn, VA) using the following parameters: A total of 3814 image stacks were acquired at a sampling rate of 1.04 Å/pixel with an 8 s exposure of 8 e$^-$ pixel$^{-1}$ s$^{-1}$ at 0.2 s/frame for a total dose of approximately 60 e$^-$ Å$^2$ s$^{-1}$ using SerialEM (*Mastronarde, 2005*).

Data was processed using cisTEM (*Grant et al., 2018*), which contains all processing steps listed below with relevant references to the technique for each step. Briefly, beam-induced motion and physical drift were corrected followed by dose-weighing using the Unblur algorithm (*Grant and Grigorieff, 2015*). Next the contrast transfer function was estimated and used to correct micrographs. After inspection of all micrographs, 3,494 were used for data processing. Particles were then automatically selected based on an empirical evaluation of maximum particle radius (80 Å), characteristic particle radius (60 Å), and threshold peak height (2 S.D. above noise). 2D classification on 842,130 particles used an input starting reference from a previous model solved in C4 symmetry (*Figure 1—*

*figure supplement 3*). This initial model was generated *ab initio* from a data set acquired for poly-cystin 2-l1 and processed using cisTEM (particles from 20 Å to 8 Å for the first step of classification with a 20 Å low pass filter used for the initial model to avoid bias). Eight class averages representing different orientations were selected for further iterative 3D classification and refinement in C4 symmetry (*Figure 1—figure supplement 3*). The best solutions for each iteration were selected for local refinement in (*Figure 1—figure supplement 3*) cisTEM. Upon completion of 3D refinement, the best class was sharpened using B-factors. Fourier Shell Correlation (FSC) and angular distribution plots were collected for the final dataset in cisTEM (*Figure 1—figure supplement 4a and b*). Final resolution maps are based on the 0.143 FSC criterion (*Rosenthal and Henderson, 2003*). 3D FSC plots were for the final data were analyzed via the web portal for 3DFSC (https://3dfsc.salk.edu) (*Tan et al., 2017*) (*Figure 1—figure supplement 4c*). Local resolution was calculated using ResMAP (*Kucukelbir et al., 2014*) (*Figure 1—figure supplement 4d*). Examples of model fit in cryo-EM maps are provided in *Figure 1—figure supplement 5* and data collection, refinment and validations statistics in *Table 1*.

**Table 1.** Cryo-EM data collection, refinement, and validation

| Data collection and processing | |
| --- | --- |
| Calibrated Magnification | 48,077 |
| Voltage (kV) | 300 |
| Electron Exposure (e-/A$^2$) | 60 |
| Defocus range (μm) | −1.0 to −2.4 |
| Pixel Size (Å/pixel) | 1.04 |
| Symmetry Imposed | C4 |
| Initial Particle images (no.) | 842,139 |
| Final Particle images (no.) | 114,814 |
| Map Resolution (Å) | 3.3 |
| FSC Threshold | 0.143 |
| Map Resolution Range | 3.3–7.1 |
| Refinement | |
| Model Resolution cutoff (Å) | 8.0 |
| FSC threshold | 0.143 |
| Map Sharpening *B* Factor (Å$^2$) | −90 |
| Model Composition | |
| Non-hydrogen Atoms | 0 |
| Protein Residues | 1656 |
| Ligands | 4 |
| R.M.S. Deviations | |
| Bond Lengths (Å) | 0.01 |
| Bond angles (°) | 1.2 |
| Validation | |
| Mol Probity Score | 1.94 |
| Clashscore | 7.6 |
| Outlier Rotamers (%) | 0.67 |
| Ramachandran Plot | |
| Favored (%) | 91.0 |
| Allowed (%) | 8.82 |
| Disallowed (%) | 0.18 |

DOI: https://doi.org/10.7554/eLife.36931.012

Polyalanine α-helices and β-sheets were built for each transmembrane section and the PMD using pdb 5T4D as a final guide in Coot (*Emsley et al., 2010*). Densities were inspected, and initial assignments of residues were based on aromatic or bulky residues. Real-space refinement, using PHENIX (*Adams et al., 2010*), was completed iteratively with inspection of the model in Coot. Model quality was evaluated using Molprobity (*Williams et al., 2018*) and EMRinger (*Barad et al., 2015*). The model was then compared to sequence alignments of the TRPP family and three models of the closely related polycystin-2 structures (*Grieben et al., 2017*; *Shen et al., 2016*; *Wilkes et al., 2017*). Pore size was evaluated using HOLE (*Smart et al., 1996*) and local resolution maps calculated from RESMAP (*Kucukelbir et al., 2014*). The sequence analysis for polycystin 2-l1 family members and projection onto the structure was completed using 500 multiple sequence alignments generated from iterative Blast searches and then analyzed with Consurf (*Glaser et al., 2003*; *Landau et al., 2005*). Electrostatic plots were created in Pymol. Model rendering was completed in Pymol and Chimera (*Pettersen et al., 2004*).

### Electrophysiology

Liposomes containing full-length polycystin 2-l1 protein were voltage clamped to record single channel currents. Protein was reconstituted following the dilution method (*Cortes and Perozo, 1997*) into liposomes made from Soy Extract Polar (Avanti). After extensive dialysis, lipids were pelleted and resuspended in HKN buffer and stored at −80°C until characterization. A sample of reconstituted protein was dried overnight at 4°C under constant vacuum and rehydrated the next morning in the same buffer. Liposomes were allowed to swell for 1 hr on ice before recording with patch pipettes (10–15 MΩ). The bath was a Tris buffered variant of HKN (in mM: TRIS 10, KCl 150, NaCl 50, $CaCl_2$ 1), pH 7.4; the pipette contained the same buffer with 1 mM $MgCl_2$. Solution osmolarity was measured at 400–405 mOsm. Data was acquired at 10 kHz and a low-pass Bessel-filtered at 1–2 kHz. For each recording, pipette offset and capacitance were corrected. Approximately 2 min after a GΩ seal was achieved, the patch was excised, and capacitance corrected before recording.

## Acknowledgements

The authors would like to extend their gratitude to members of the Clapham and Grigorieff labs, and the staff of the Cryo-EM facility at Janelia Research Campus for their feedback and support.

## Additional information

### Funding

| Funder | Author |
| --- | --- |
| Howard Hughes Medical Institute | David E Clapham |

The funders had no role in study design, data collection and interpretation, or the decision to submit the work for publication.

### Author contributions

Raymond E Hulse, Conceptualization, Data curation, Formal analysis, Supervision, Validation, Investigation, Visualization, Methodology, Writing—original draft, Writing—review and editing; Zongli Li, Rick K Huang, Resources, Investigation, Writing—review and editing; Jin Zhang, Investigation, Methodology, Writing—review and editing; David E Clapham, Conceptualization, Supervision, Funding acquisition, Writing—original draft, Project administration, Writing—review and editing

### Author ORCIDs

Raymond E Hulse http://orcid.org/0000-0002-0110-3752
David E Clapham http://orcid.org/0000-0002-4459-9428

### Decision letter and Author response
Decision letter https://doi.org/10.7554/eLife.36931.025

Author response https://doi.org/10.7554/eLife.36931.026

## Additional files

### Supplementary files
• Transparent reporting form
DOI: https://doi.org/10.7554/eLife.36931.013

### Data availability
The cryo map and model have been deposited to the Worldwide Protein Data Bank (6DU8) and Electron Microscopy Data Bank (8912).

The following datasets were generated:

| Author(s) | Year | Dataset title | Dataset URL | Database, license, and accessibility information |
|---|---|---|---|---|
| Clapham DE, Hulse RE, Li Z, Huang RK, Zhang J | 2018 | Human Polycystin 2-l1 | https://www.rcsb.org/structure/6DU8 | Publicly available at the RCSB Protein Data Bank (Accession no: 6DU8) |
| Clapham DE, Hulse RE, Li Z, Huang RK, Zhang J | 2018 | Human Polycsytin 2-l1 | http://www.ebi.ac.uk/pdbe/entry/emdb/EMD-8912 | Publicly available at the Electron Microscopy Data Bank (Accession no: 8912) |

The following previously published datasets were used:

| Author(s) | Year | Dataset title | Dataset URL | Database, license, and accessibility information |
|---|---|---|---|---|
| Shen PS, Yang X, DeCaen PG, Liu X, Bulkley D, Clapham DE, Cao E | 2016 | Cryo-EM structure of Polycystic Kidney Disease protein 2 (PKD2), residues 198-703 | https://www.rcsb.org/structure/5T4D | Publicly available at the RCSB Protein Data Bank (accession no: 5T4D) |
| Wilkes M, Madej MG, Ziegler C | 2016 | cryoEM Structure of Polycystin-2 in complex with calcium and lipids | https://www.rcsb.org/structure/5MKF | Publicly available at the RCSB Protein Data Bank (accession no: 5MKF) |
| Wilkes M, Madej MG, Ziegler C | 2016 | cryoEM Structure of Polycystin-2 in complex with cations and lipids | https://www.rcsb.org/structure/5MKE | Publicly available at the RCSB Protein Data Bank (accession no: 5MKE) |

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
