## [Decision Letter]

Thank you for submitting your article "Cryo-EM structure of the polycystin 2-l1 ion channel" for consideration by *eLife*. Your article has been reviewed by 3 peer reviewers, and the evaluation has been overseen by Richard Aldrich as the Senior Editor, a Reviewing Editor, and three reviewers. The reviewers have opted to remain anonymous.

The reviewers have discussed the reviews with one another and the Reviewing Editor has drafted this decision to help you prepare a revised submission.

Summary:

In this study, Hulse et al., presented a structure of the full-length human polycystin 2-l1 protein at 3.1Å resolution using single particle electron cryo-microscopy (cryo-EM). Although it shares high sequence similarity to polycystin 2, whose structure has been determined by multiple labs recently, polycystin 2-l1 exhibits distinct ion permeability. The study aims to provide a detailed view of the structural differences between polycystin 2-l1 and polycystin 2, and these structural differences may underlie the functional difference between the two channels.

Essential revisions:

1) The manuscript is structured from the viewpoint of understanding the function of the channel but, at present, it is not clear at this point whether the structural differences contribute to function. The main impact of the present study would be to compare the structures of polycystin 2-l1 and PKD2 in as much detail as possible. Oddly, the introduction to the manuscript barely mentions the fact that a structure for PKD2 has already been reported by this and other groups, but the trajectory of the manuscript would be better served by addressing this up front and moving on from there. In this context, figures showing direct comparisons or overlays of these two TRPP structures would be very useful and help convey important points that are difficult to ascertain from the current format. We are not expecting new functional data but the new structures at the least should help develop a cogent hypothesis for further testing.

2) There seems to be a disconnect between the stated resolution of the structure and the quality of the maps. This is suggested by the inability to resolve many regions and, in particular, side chains. From the figures provided (for example, Figure 2B and Figure 3D), it is not clear how well the model fits the density map in various regions. The maps look more to be in the 4 – 4.5Å range, and it isn't clear where the possible discrepancy lies. Overfitting is a known issue with cisTEM program for EM structure determination. The authors should re-evaluate their data processing and analysis.

3) Related to the above point, please provide the map for this structure which would help reviewers assess the quality of the data and thus the validity of the conclusions.

4) Another question concerns the unusual protocol for channel protein purification, which is very different from that described for PKD2 and which involves initial extraction into C12E9 detergent, batch overnight incubation with amylose resin, and multiple gel filtration steps. Could this be enhancing protein instability that degrades resolution?

5) Overall, the manuscript is difficult to follow. Most figures are not properly labeled and many of the descriptions in the paper are not adequately supported by figures, making it difficult to comprehend, particularly for readers who are not familiar with the TRPP family of channels. Specifically:

Results section: "We conclude that these elements are either unstructured or connected with flexible regions, which, along with the lack of cytoplasmic elements in 2D classification (Shen et al., 2016), prevents model building of these regions." This sentence is unclear and I am not sure what the phrase "cytoplasmic elements" refer to.

Subsection “*The Voltage Sensing-Like Domain*”: "This S3 helix exhibits greater secondary structure than the presumed closed polycystin-2 structure of Shen et al." What does greater secondary structure mean here – longer S3?

Subsection "The Polycystin Mucolipin Domain". The authors spend so much effort pointing out very detailed structural differences between polycystin 2-l1 and polycystin 2. However, these differences do not seem to have any functional relevance.

Subsection “Three leaf clovers of the polycystin mucolipin domain”: "However, the nature of the interaction differs in that F216 of TLC1 is near W308 of the adjacent subunit's PMD, representing a possible pi-pi stacking interaction or a hydrophobic pocket". A hydrophobic pocket is not a proper way to describe this interaction.

Subsection “Fenestrations and TLC3”: If I recall correctly, the fenestration appears to be sealed in polycystin-2 by an extended loop based on Shen's structure. If there is a fenestration in the present structure, a proper figure should be provided here.

Subsection “Fenestrations and TLC3”: What does "upper pore domain" refer to: the extracellular side of the pore domain? What do "the loop pore helices" refer to here: the loop and/or the pore helices? What does "mutually exclusive glycosylation states" refer to? In the last sentence of this section: "This difference may help rationalize the intermediate diameter observed in the polycystin-2 multiple-ion structure (1.4Å) (Wilkes et al., 2017) and the single-ion structure (1.0Å) (Wilkes et al., 2017) where such interactions do not exist, and our polycystin 2-l1 structure (3.1Å)." What diameters are the authors describing here and what does "this difference" refer to? Again, there is no figure to support all the descriptions in this paragraph.

Subsection “The Pore Domain”:

I am not sure how the figures are ordered. Figure 2 is described after Figure 4. Also, there are no proper labels on Figure 2.

First paragraph of subsection “The Pore Domain”: "Notably, one element, K511 in polycystin 2-l1, is highly variable among TRPPs." Where is this K511? Again, there is no figure to support it.

Second paragraph subsection “The Pore Domain”: I simply don't understand what is discussed here and again there is no figure to support the description. Should the filter diameter be radius? Does 5K47 refer to a PDB code?

Third paragraph of subsection “The Pore Domain”: The narrowest distance of 2.1 Å appears to be the narrowest filter radius based on Figure 2A (right panel), which excludes the van deer Waals radius of the G522 carbonyl oxygen. The "2.4 Å metal-oxygen distance" refers to atom-to-atom distance between Ca^2+^ and oxygen. These two distances are not comparable.

Third paragraph of subsection “The Pore Domain”: Here the author repeated the same discussion as the last paragraph, except using atom-to-atom diagonal distances. If the diagonal distance between the two G522 carbonyl oxygen atoms is 8.7Å, then the minimum distance between the G522 carbonyl oxygen and Ca^2+^ ion would be 4.4 Å. If so, the argument of 2.4 Å metal-oxygen distance is no longer correct. Basically, the whole description about the pore domain is messy and incomprehensible.

Subsection “Ion permeation differences between polycystin-2 and polycystin 2-l1”. The values of relative permeability ratio should be provided.

"Higher resolution Cryo-EM structures that identify Ca^2+^ and/or monovalents that are occupying the pore with high certainty (or equivalent crystallographic studies like those of CaVAb (Tang et al., 2014)), and realistic molecular dynamic simulations, should shed light on these issues." What is realistic MD simulation? What do "these issues" refer to here?

I simply don't understand the logic behind the electrostatic argument. Are the authors suggesting that the net positive charge of the polycystin 2-l1 pore helices contributes to the Ca^2+^ permeability of the channel? Also, how do they compare the relative Ca^2+^ permeability among four different channels?

Discussion section "The relatively larger diameter of polycystin 2-l1's pore (0.6Å – 1.1Å) may enable partially […]" Where is this distance value from? It does not match with the pore radius plot! Is it 0.6-1.1 Å larger than the pore of polycystin 2?

Figure 1A the pore radius and the solvent accessible pathway does not match, particularly the lower gate (green) part.

---

## [Author Response]

Essential revisions:1) The manuscript is structured from the viewpoint of understanding the function of the channel but, at present, it is not clear at this point whether the structural differences contribute to function. The main impact of the present study would be to compare the structures of polycystin 2-l1 and PKD2 in as much detail as possible. Oddly, the introduction to the manuscript barely mentions the fact that a structure for PKD2 has already been reported by this and other groups, but the trajectory of the manuscript would be better served by addressing this up front and moving on from there. In this context, figures showing direct comparisons or overlays of these two TRPP structures would be very useful and help convey important points that are difficult to ascertain from the current format. We are not expecting new functional data but the new structures at the least should help develop a cogent hypothesis for further testing.

We have taken your point and extensively discuss the PKD2/PKD-2l1 structures. We believe that the most relevant comparisons are made in Figure 5 with PKD2 and TRPV5 and TRPV6. We emphasize that despite PDK2’s similarity with PKD2L1 (structural and sequence), there are differences in Ca^2+^ permeation physiologically: polycystin-2l1 is permeant to calcium, but polycystin-2 is not. We have included a comparison of the profile of the selectivity filter and mapped the RMSD difference onto the structure as well as provided an overlay (Figure 1—figure supplement 5).

*2) There seems to be a disconnect between the stated resolution of the structure and the quality of the maps. This is suggested by the inability to resolve many regions and, in particular, side chains. From the figures provided (for example, Figure 2B and Figure 3D), it is not clear how well the model fits the density map in various regions. The maps look more to be in the 4 – 4.5*Å *range, and it isn't clear where the possible discrepancy lies. Overfitting is a known issue with cisTEM program for EM structure determination. The authors should re-evaluate their data processing and analysis.*

We agree. Further evaluation of the data after consultation with Niko Grigorieff’s group revealed that the data was slightly overfit. To address this concern, we reviewed our data processing strategy and now have limited the resolution during refinement to 8Å. This decreased the nominal resolution but improved map density, with associated improved 3D FSCs (see below). We’ve re-evaluated the data with new tools from the Lyumkis group using 3D FSC plots to characterize the preferred orientation observed in our data Tan et al., 2017. The 3D FSC results are consistent with the angular distribution plot (Figure 1—figure supplement 4) which show that while the Z-resolution is high (labeled side view), the in-plane (x/y) resolution is weaker. This is also reflected in the fits of side chains in the map such as elements seen in Figure 1—figure supplement 5B. We have included the results of this analysis as well as new maps and models. We hope that our analysis and the new tools will help other ion channel groups facing similar problems with their structural efforts using cryoEM. While we cannot eliminate the preferred orientation of our channel, we believe that Lyumkis’ methods enable a better understanding of the effect on the data. Additional metrics, such as the fit of the map (Figure 1—figure supplement 5), the high degree of similarity to Erhu Cao’s polycystin-2 model (Figure 1—figure supplement 2), and the similarity to the mouse polycystin 2-l1 model that was released very shortly after we submitted this work (Su et al., 2018) all verify that our model is not significantly affected. The current estimation from cisTEM and the 3D FSC is 3.3Å and 3.2Å respectively. We report our resolution at the more conservative 3.3Å. The model was reevaluated with the new maps, and several minor adjustments were made including the ability to build in a side chain for residue D523, a critical component of the selectivity filter. We also reveal a more complete loop in the PMD, and some improved density fitting at linkers between transmembrane helices. All figures and the manuscript were updated with this information.

3) Related to the above point, please provide the map for this structure which would help reviewers assess the quality of the data and thus the validity of the conclusions.

Done.

4) Another question concerns the unusual protocol for channel protein purification, which is very different from that described for PKD2 and which involves initial extraction into C12E9 detergent, batch overnight incubation with amylose resin, and multiple gel filtration steps. Could this be enhancing protein instability that degrades resolution?

Our protocol is based on empirical evaluation of extraction, stabilization and exchange using standard techniques. C12E9 was selected as an efficient extraction detergent on an empirical screen using western blot against MBP. Stability is evident from the exchange into DDM:CHS during amylose chromatography and in the SEC (Figure 1—figure supplement 1) and the ability of the protein to remain in solution for up to 72 h with stable SEC profiles (not shown).

5) Overall, the manuscript is difficult to follow. Most figures are not properly labeled and many of the descriptions in the paper are not adequately supported by figures, making it difficult to comprehend, particularly for readers who are not familiar with the TRPP family of channels. Specifically:Results section: "We conclude that these elements are either unstructured or connected with flexible regions, which, along with the lack of cytoplasmic elements in 2D classification (Shen et al., 2016), prevents model building of these regions." This sentence is unclear, and I am not sure what the phrase "cytoplasmic elements" refer to.

Polycystin 2-l1’s C terminus is cytoplasmic, likely forming a coiled-coiled motif like other homotetrameric channels. In Shen, 2016, a truncation of the C terminus was required for a stable biochemical preparation. We were able to express full-length protein, but the C terminus was still unresolvable

Subsection “The Voltage Sensing-Like Domain”: "This S3 helix exhibits greater secondary structure than the presumed closed polycystin-2 structure of Shen et al." What does greater secondary structure mean here – longer S3?

Yes, S3 in polycystin 2-l1 has more α helical content and is thus continuously longer than the interrupted C terminus of Shen et al., 2016. We have clarified the text.

Subsection "The Polycystin Mucolipin Domain". The authors spend so much effort pointing out very detailed structural differences between polycystin 2-l1 and polycystin 2. However, these differences do not seem to have any functional relevance.

We believe that the PMD will turn out to be a ligand binding domain, but to date, none has been found (for the PMD of any type II TRP). Since disruption of the PMD results in nonfunctional channels, it will take some time to understand its function.

Subsection “Three leaf clovers of the polycystin mucolipin domain”: "However, the nature of the interaction differs in that F216 of TLC1 is near W308 of the adjacent subunit's PMD, representing a possible pi-pi stacking interaction or a hydrophobic pocket". A hydrophobic pocket is not a proper way to describe this interaction.

We have corrected the manuscript.

Subsection “Fenestrations and TLC3”: If I recall correctly, the fenestration appears to be sealed in polycystin-2 by an extended loop based on Shen's structure. If there is a fenestration in the present structure, a proper figure should be provided here.

The fenestration seen in polycystin 2-l1 is a small hole at the interface of the top of the VSLD and bottom of the PMD. Polycystin-2 has a similar configuration, with a small hole (5T4D). Whether the PMD is truly sealing this entry is unclear in our model since we can only approximate the position relative to the membrane.

Subsection “Fenestrations and TLC3”: What does "upper pore domain" refer to: the extracellular side of the pore domain? What do "the loop pore helices" refer to here: the loop and/or the pore helices? What does "mutually exclusive glycosylation states" refer to? In the last sentence of this section: "This difference may help rationalize the intermediate diameter observed in the polycystin-2 multiple-ion structure (1.4Å) (Wilkes et al., 2017) and the single-ion structure (1.0Å) (Wilkes et al., 2017) where such interactions do not exist, and our polycystin 2-l1 structure (3.1Å)." What diameters are the authors describing here and what does "this difference" refer to? Again, there is no figure to support all the descriptions in this paragraph.

The comparison of polycystin-2 and polycystin 2-l1 in the supplementary section helps to clarify this discussion.

Subsection “The Pore Domain”:I am not sure how the figures are ordered. Figure 2 is described after Figure 4. Also, there are no proper labels on Figure 2.First paragraph of subsection “The Pore Domain”: "Notably, one element, K511 in polycystin 2-l1, is highly variable among TRPPs." Where is this K511? Again, there is no figure to support it.Second paragraph of subsection “The Pore Domain”: I simply don't understand what is discussed here and again there is no figure to support the description. Should the filter diameter be radius? Does 5K47 refer to a PDB code?Third paragraph of subsection “The Pore Domain”: The narrowest distance of 2.1 Å appears to be the narrowest filter radius based on Figure 2A (right panel), which excludes the van deer Waals radius of the G522 carbonyl oxygen. The "2.4 Å metal-oxygen distance" refers to atom-to-atom distance between Ca^2+^ and oxygen. These two distances are not comparable.

We apologize for the confusion. Our attempt to discuss both the radii of the measurements from the HOLE program (which calculates the VDWr for each atom, with a spherical probe for our case) and measurements from Pymol (as either Cα or CO distances) present different information. HOLE is better suited to this discussion. We’ve removed the Pymol distances. We’ve also corrected the specific questions, improved the figure legends and ensured the order is correct.

Third paragraph of subsection “The Pore Domain”: Here the author repeated the same discussion as the last paragraph, except using atom-to-atom diagonal distances. If the diagonal distance between the two G522 carbonyl oxygen atoms is 8.7Å, then the minimum distance between the G522 carbonyl oxygen and Ca^2+^ ion would be 4.4Å. If so, the argument of 2.4Å metal-oxygen distance is no longer correct. Basically, the whole description about the pore domain is messy and incomprehensible.Subsection “Ion permeation differences between polycystin-2 and polycystin 2-l1”. The values of relative permeability ratio should be provided."Higher resolution Cryo-EM structures that identify Ca^2+^ and/or monovalents that are occupying the pore with high certainty (or equivalent crystallographic studies like those of CaVAb (Tang et al., 2014)), and realistic molecular dynamic simulations, should shed light on these issues." What is realistic MD simulation? What do "these issues" refer to here?I simply don't understand the logic behind the electrostatic argument. Are the authors suggesting that the net positive charge of the polycystin 2-l1 pore helices contributes to the Ca^2+^ permeability of the channel? Also, how do they compare the relative Ca^2+^ permeability among four different channels?

We have added the relative permeabilities from previous lab work (DeCaen, 2013) and cited literature (Ramsey, 2006).

Yes, we believe that the overall difference in charge at this region contributes to the moderate increase in Ca^2+^ relative to polycystin-2.

Polycystin-2 is nonselective among monovalents and does not conduct calcium, whereas polycystin 2-l1 is nonselective among monovalens but also conducts Ca^2+^ (with weak selectivity over Na^+^). TRPV5 and TRPV6 have a much higher selectivity (>10x) for Ca^2+^ over Na^+^ than polycystin 2-l1. We believe that both the local energy landscape and geometry of the selectivity filter are at least two variables that distinguish polycystin 2-l1 from other TRPP and TRPV channels.

Discussion section "The relatively larger diameter of polycystin 2-l1's pore (0.6Å – 1.1Å) may enable partially […]" Where is this distance value from? It does not match with the pore radius plot! Is it 0.6-1.1 Å larger than the pore of polycystin 2?

This was an error and has been fixed.

Figure 1A the pore radius and the solvent accessible pathway does not match, particularly the lower gate (green) part.

This figure was updated with the new model.